# 1-Year Mortality Prediction through Artificial Intelligence Using Hemodynamic Trace Analysis among Patients with ST Elevation Myocardial Infarction

**DOI:** 10.3390/medicina60040558

**Published:** 2024-03-29

**Authors:** Seyed Reza Razavi, Tyler Szun, Alexander C. Zaremba, Ashish H. Shah, Zahra Moussavi

**Affiliations:** 1Biomedical Engineering Program, University of Manitoba, Winnipeg, MB R3T 5V6, Canada; razavisr@myumanitoba.ca; 2Department of Medicine, Rady Faculty of Health Sciences, University of Manitoba, Winnipeg, MB R3E 3P5, Canada; szunt@myumanitoba.ca (T.S.); zarembaa@myumanitoba.ca (A.C.Z.); ashah5@sbgh.mb.ca (A.H.S.)

**Keywords:** ST elevation myocardial infarction (STEMI), mortality, arterial pressure, machine learning, classification

## Abstract

*Background and Objectives*: Patients presenting with ST Elevation Myocardial Infarction (STEMI) due to occlusive coronary arteries remain at a higher risk of excess morbidity and mortality despite being treated with primary percutaneous coronary intervention (PPCI). Identifying high-risk patients is prudent so that close monitoring and timely interventions can improve outcomes. *Materials and Methods*: A cohort of 605 STEMI patients [64.2 ± 13.2 years, 432 (71.41%) males] treated with PPCI were recruited. Their arterial pressure (AP) wave recorded throughout the PPCI procedure was analyzed to extract features to predict 1-year mortality. After denoising and extracting features, we developed two distinct feature selection strategies. The first strategy uses linear discriminant analysis (LDA), and the second employs principal component analysis (PCA), with each method selecting the top five features. Then, three machine learning algorithms were employed: LDA, K-nearest neighbor (KNN), and support vector machine (SVM). *Results*: The performance of these algorithms, measured by the area under the curve (AUC), ranged from 0.73 to 0.77, with accuracy, specificity, and sensitivity ranging between 68% and 73%. Moreover, we extended the analysis by incorporating demographics, risk factors, and catheterization information. This significantly improved the overall accuracy and specificity to more than 76% while maintaining the same level of sensitivity. This resulted in an AUC greater than 0.80 for most models. *Conclusions*: Machine learning algorithms analyzing hemodynamic traces in STEMI patients identify high-risk patients at risk of mortality.

## 1. Introduction

Myocardial infarction (MI), or a heart attack, is one of the leading causes of death worldwide [1,2]. It is estimated to result in over 4 million deaths in Europe and northern Asia and 2.4 million deaths in the United States each year [3]. In 2022, heart disease ranked as the second leading cause of death in Canada [4]. Moreover, in the USA alone, approximately USD 29.8 billion was spent on the direct management of MI in 2016 [5].

In current medical practice, MI is identified based on clinical presentation, dynamic electrocardiogram (ECG) changes, and a rise in troponin, a cardiac-specific biomarker. Patients exhibiting ST-segment elevation with reciprocal changes in ECG are diagnosed with ST elevation myocardial infarction (STEMI) [6]; that is typically because of complete occlusion of coronary arteries [7]. In comparison, patients with chest pain, a rise in troponin levels, and ECG changes other than STEMI are defined as having non-STEMI (NSTEMI). Primary percutaneous coronary intervention (PPCI) is the gold-standard care for patients with STEMI. However, this procedure is time sensitive. Hence, patients who cannot be brought to a cardiac catheter laboratory within 120 min of their first contact with medical personnel are treated with thrombolytic therapy aiming at dissolving intracoronary clots, restoring flow, and transferring them to the nearby cardiac center for further care.

Mortality and morbidity among MI patients have markedly improved over the last four decades, primarily due to proactive detection and management of cardiovascular risk factors [8], along with timely myocardial salvage by coronary revascularization [9]. Despite such success, 30-day mortality among patients admitted with MI remains between 6.5–9.3% across the European countries [10]. Data from Denmark, where all STEMI patients are treated with PPCI, demonstrated a 1-year mortality reduction from 10.8% (2003–2006) to 7.7% (2015–2018); the majority of this mortality reduction was observed within the first 30 days [11]. In addition to higher mortality, these patients may also experience complications, including life-threatening arrhythmias, heart failure, a prolonged in-hospital stay, and various mechanical complications despite successful PPCI [12,13,14]. Hence, it is prudent to identify high-risk patients who may benefit from close monitoring and timely intervention that may plausibly improve their outcomes.

Various risk assessments in the context of myocardial infarction (MI) have been developed. The Global Registry of Acute Coronary Events (GRACE) [15], the most widely used source that is recommended by the European Society of Cardiology STEMI guidelines [16], estimates the mortality risk in hospital, at 6 months, 1 year, and 3 years. The thrombolysis in myocardial infarction (TIMI) [17] risk score was initially developed for 30-day mortality in patients after thrombolysis and then validated for patients with STEMI [18]. Based on clinical and electrocardiographic characteristics, the primary angioplasty in myocardial infarction (PAMI) score is used to predict late mortality in patients with STEMI treated by PPCI [19]. The controlled used of abciximab and the investigation of device usage to lower late angioplasty complications (CADILLAC) considers the initial measurement of left ventricular function and predicts 1-year mortality [20]. Finally, the Zwolle [21] score was developed for 30-day mortality prediction.

These traditional methods for determining cardiovascular disease (CVD) risk typically presuppose a linear relationship between risk factors and clinical outcomes. However, such a linear approach might be oversimplifying their relationship. Cardiovascular diseases are inherently complex and diverse, influenced by genetic predispositions, environmental conditions, and lifestyle choices [16,22]. These approaches primarily focus on conventional prognostic factors [23], limiting their effectiveness due to the emerging need to incorporate and examine various information sources, including those describing MI-related pathophysiology. Moreover, these scoring systems are routinely not utilized in the current era of prompt coronary revascularization. Aortic pulse wave is a physiological marker describing cardiovascular health [24,25] that may provide valuable information about changing physiological status among patients undergoing PPCI. 

Machine learning (ML) has the potential to bypass the restrictions of the approaches mentioned above [26]. Static assumptions about data behavior do not constrain ML data analysis and can train models to uncover patterns within the data. The application of ML, especially in predicting in-hospital mortality [27], 30-day mortality [28], short- and long-term mortality [29], arrhythmia [30], and readmission [31] has seen rapid growth. ML has been widely compared with traditional methods such as TIMI and GRACE. ML has demonstrated superior performance to traditional risk-scoring methods in mortality prediction [32,33,34,35,36]. ML outperformed the TIMI score in predicting both short- and long-term mortality [33]. Additionally, it demonstrated better outcomes for 30-day [35] and 1-year [32,34,36] mortality predictions compared to the GRACE score for patients with STEMI. Most ML algorithms have primarily employed continuous and categorical data from patients’ records during angioplasty. To the best of our knowledge, none of the previous ML research has focused on extracting features from hemodynamic traces, such as the arterial pressure (AP) signal obtained throughout the PPCI procedure.

## 2. Materials and Methods

### 2.1. Study Population and Data Acquisition

This retrospective study included 800 consecutive patients suspected to have STEMI who were referred to the St Boniface General Hospital, Winnipeg, MB, Canada, for consideration of PPCI between January 2020 and October 2021.

Patients’ demographics, cardiovascular risk factors, catheterization data and outcomes were collected by reviewing individual electronic patient records (EPRs). The arterial pressure (AP) wave tracings throughout the PPCI procedure were obtained through the MacLab system database (GE Healthcare; Milwaukee, WI, USA). Given these retrospective data analyses, individual patient consent was not obtained. This study was approved by the local Research and Ethics Board, University of Manitoba [REB: HS25542 (H2022: 196)].

During catheterization, specifically the pullback of a catheter from the left ventricle (LV) back to the ascending aorta (AO) across the aortic valve, the MacLab software calculates the ejection systolic period (ESP), which is the duration of ejection in seconds/minute that can be converted to left ventricular ejection systolic time (EST) measured in seconds/heartbeat.

### 2.2. Statistical Analysis

Statistical analysis was performed using MATLAB software 9.13. The mean values were compared using the Mann–Whitney U test for continuous variables. In contrast, categorical variables were analyzed using the χ2 (chi-square) test. A two-tailed *p*-value of less than 0.05 was considered statistically significant. Effect size was calculated using the phi coefficient (φ) for categorical variables and Cohen’s d for continuous variables. Effect size measures the strength and practical significance of a relationship or difference between variables, showing the extent of this variation or association in real-world contexts. Effect sizes of 0.10, 0.30, and 0.50 indicate small, medium, and large effects, respectively [37].

### 2.3. Pre-Processing and Denoising of Data

Our primary objective is to derive features from aortic pulse wave tracing obtained throughout the PPCI procedure. However, such AP signals can be affected by different types of noise. This includes motion artifacts (resulting from the movement of the catheter, transducer, or patient), electrode polarization, and electrical interference. Figure 1a illustrates an AP signal captured using MacLab that requires noise reduction. Figure 2 displays different parts of a typical recorded AP signal. As can be seen in Figure 1b, the selected part is noisy and should be excluded from the analysis. On the other hand, Figure 1c illustrates a part of the signal suitable for further analysis and feature extraction. This shows the importance of having a denoising strategy to extract the high-quality part of the AP signal. 

The block diagram of the denoising procedure is shown in Figure 2. It can generally be summarized as windowing the AP signal and discarding the noisy windows. We extracted the heartbeat (HB) from lead II ECG (also recorded by MacLab) to achieve an adaptive window size. We considered the window size to be six times the heart cycle length of each individual’s data. We extracted the heart rate using the Pan–Tompkins algorithm [38]. This algorithm employs band-pass filtering to enhance the signal-to-noise ratio (SNR) and eliminate low-frequency artifacts. A derivative operation is used to diminish the P and T waves, thus highlighting the QRS complex. This is followed by a squaring operation that amplifies the high-frequency elements. Subsequently, moving window integration is applied to create a smooth pulse corresponding to each QRS complex. Finally, the R peaks in QRS complexes are identified through adaptive thresholding, as depicted in Figure 3. 

After extracting the HB, we windowed the high-pass-filtered AP signal (with a cutoff frequency of 0.4 Hz) using a 50% overlap and a duration of six heart cycles. Following the windowing of the AP signal, we computed the fractal dimension (FD), mean, and standard deviation (SD). In this research, we used one of the most commonly used algorithms to estimate the fractal dimension: the Katz fractal dimension (KFD) [39]. The comprehensive set of features derived from each window of the AP signal includes: Mean values: Each window was segmented into three equal parts. For each segment, the mean value of the AP signal was calculated, resulting in three values.SD analysis: This involves calculating four SD values. The SD was computed for the entire window, and then the window was divided into three equal parts to determine the SD for each segment.FD Calculation: Each window was divided into three segments, and the FD was computed for each segment.

These characteristics were calculated for every AO window. To eliminate noisy windows, we applied thresholds to each characteristic, which were determined based on the physiological properties of the AO signal.

### 2.4. Feature Extraction

We separated each AP waveform after denoising the AP signal and discarding the noisy AP windows. From each waveform, we extracted a total of 18 features, as detailed in Table 1. The first 14 features relate to the time aspect, whereas spectral entropy (SE) and average power (Pave) are associated with frequency. We have tried to capture every aspect of the AP that might be useful for prediction. In the table below, skewness quantifies the asymmetry in a data set, while kurtosis evaluates whether the data have heavier or lighter tails compared to a normal distribution. SE measures the irregularity or complexity of digital signals within the frequency domain, and Pave indicates the mean energy transmission of a signal over a certain period.

Except for the overall time (OT) feature, which refers to the entire duration of the surgery, all other extracted features have multiple values per subject. This is because they are extracted from each AP waveform, and we have multiple waveforms for each subject’s AP signal. We applied a 20% trimmed mean for multi-value features to derive a single representative value for each characteristic per subject. The trimmed mean method excludes a specified percentage of the extreme values, both largest and smallest, before the mean calculation. This technique is beneficial in reducing the impact of outliers that could potentially bias the traditional mean.

### 2.5. Feature Selection

We also implemented feature selection methods to reduce the number of variables further, thereby shortening training time and enhancing model performance. We employed two feature-reduction methods: principal component analysis (PCA) [40] and linear discriminant analysis (LDA) [41]. In both approaches, we selected the top five features.

#### 2.5.1. LDA-Based Feature Selection (LBFS)

LDA, a supervised learning algorithm, was used for the feature selection in machine learning. As described in Figure 4, for each feature we utilized an individual LDA classifier to determine the power of each feature in distinguishing two classes. We divided our dataset into two sections (70% for training and 30% for validation) and classified patients into two groups: survivors and non-survivors at one year after admission. After training, we evaluated the sensitivity of each classifier, focusing on the top five features with the highest sensitivity. This emphasis on sensitivity is crucial for accurately predicting the non-surviving group, a key concern in our research.

This procedure was iterated 1000 times, with the training and validation sets being reshuffled each time. To tackle issues related to class imbalance, we employed the technique of down sampling (we randomly selected from class 1 (survived) to attain a more balanced number of patients across both classes).

#### 2.5.2. PCA-Based Feature Selection (PBFS)

PCA is a widely recognized technique used for extracting features and reducing dimensionality. PCA aims to project data that initially existed in a d-dimensional space into a space of lower dimensions. In PCA, the process begins with calculating the dataset’s mean vector and covariance matrix. Then, eigenvectors and eigenvalues are computed and sorted by the eigenvalues’ magnitude. The largest k eigenvectors are selected, often based on an eigenvector spectrum analysis. The PCA output is a k-vector that prioritizes significance, with the first few principal components usually representing most of the dataset’s variability.

To employ PCA for feature selection, we initially calculated PCA using all features. Subsequently, we focused on the first four eigenvectors (PCA 1–PCA 4), which together represent nearly 80 percent of the variability in the dataset. A limitation of these PCAs is their lack of clarity regarding which specific features contribute most significantly to their formation, thereby not clearly indicating the most important features. To address this, we calculated the correlation between each feature and these four PCA vectors (Figure 5). We then computed the average of the values in each column, resulting in a single vector. The next step was to identify the feature corresponding to the highest value in this vector. We selected the five top features with higher correlation as the features selected by this method.

Like the previous feature selection method, we addressed the imbalance issue by using the down-sampling method and repeating the process 1000 times to avoid bias towards a particular class. Additionally, we performed scaling before conducting the PCA. The block diagram of the implemented PBFS is shown in Figure 5.

### 2.6. Data Imbalance and Separation of Data into Training and Testing

The effectiveness of many standard binary classification algorithms in machine learning is higher with balanced datasets, as highlighted in [42]. However, the true challenge arises with imbalanced datasets, where these algorithms often struggle, especially since the consequences of misclassifying the minority class tend to be significantly more severe than those of misclassifying the majority class. Numerous strategies have been developed to manage imbalanced datasets. However, these methods have been criticized for changing the dataset’s original class distribution by generating new data (over-sampling), which may lead to overfitting, or by removing valuable data (under-sampling).

Our proposed ensemble-based methods will try to tackle the possible problems of the conventional methods mentioned above for handling class imbalance problems by converting an imbalanced dataset into several balanced datasets that do not suffer anymore from the challenge of an imbalanced dataset without creating new extra data or discarding potentially useful original data. As depicted in Figure 6, our method utilized a 10-fold cross-validation technique for the entire dataset. This ensured that the distribution of the whole data set was maintained for our test set.

We used an ensemble method for the training part of the study: We created ‘*n*’ balanced sub-datasets from the imbalanced training dataset. To form these sub-datasets, we included all subjects from the minority class and randomly selected an equal number from the majority class. After selection, these subjects were removed from the majority dataset. Each sub-dataset, starting with the first, was used to train a model (referred to as ‘algo 1’ in Figure 6). We repeated this process across the entire majority dataset, resulting in ‘n’ distinct models. This approach ensured that no subjects were discarded, thus preserving essential information. Due to the disparity in the number of subjects between the minority and majority classes, our finally created sub-dataset might be slightly imbalanced. After training the models, we used the test set and evaluated each model’s performance and averaged the results of all the trained models. Eventually, because we implemented the 10-fold technique, we achieved the performance of that fold each time. In the end, we also averaged the test results from all of the folds and reported this value.

### 2.7. ML Predictive Models

The predictive models for 1-year mortality were developed using three different machine learning techniques: K-nearest neighbor (KNN) [43], LDA [41], and support vector machine (SVM) [44]. The KNN classifier, used for multiclass classification, identifies the nearest neighbors by calculating distances between a test sample and training data. KNN determines the nearest neighbors and uses a majority vote among them to classify the new sample. SVM, notable in biomedical fields for its precision and handling of multiple predictors [45], works by finding an optimal hyperplane for linear separation between classes. It categorizes data using this hyperplane, and it is effective in linear and nonlinear contexts. LDA combines features into a new variable to distinguish classes in a dataset. It reduces multi-dimensional data to one dimension, aiming for distinct class separation based on the discriminant score. This simplifies analysis and highlights differences between classes.

These models were trained and tested using two different strategies. Initially, we trained and tested the three models mentioned above using the top five features selected by each of our proposed feature selection methods. This evaluated the impact of considering only the top five features derived from the AP signal. In the second part, we explored the potential of enhancing model performance by incorporating demographic, risk factor, and catheterization data. These additional features were added to the top five AP curve features that were selected based on their *p*-value and effect size.

### 2.8. Models’ Evaluation

Evaluating the effectiveness of machine learning algorithms is crucial for determining their performance. We assessed our proposed machine learning approaches using accuracy, specificity, sensitivity (recall), and precision measures. Additionally, we plotted the ROC (receiver operating characteristic) curve to demonstrate the performance of our binary classification model at various thresholds. Subsequently, we computed the AUC (area under the ROC curve), a singular metric summarizing the overall effectiveness of the binary classification model.

## 3. Results

### 3.1. Patients’ Characteristics

Out of 800 patients, data from 605 patients [age: 64.2 + 13.2 years; 432 (71.41%) males] with STEMI were selected for analysis. In total, 195 patients were excluded due to lacking adequate AP signals (*n* = 17), having poor-quality signals (*n* = 14), having an alternative diagnosis other than STEMI (*n* = 118), or not having catheterization data (*n* = 46). Figure 7 shows this exclusion schematically. The included patients were identified as having STEMI through ERP. This study reports 1-year mortality, defined as the period starting from admission, with patient follow-ups confirming the outcomes.

Among these patients, STEMI localization was inferior [313 (51.74%)], anterior [230 (38.02%)], lateral [43 (7.11%)] and posterior [19 (3.14%)]. We investigated how the site of STEMI impacted patient survival, differentiating between survivors and non-survivors. Our research found no significant connection between survival rates and the STEMI’s location (inferior, anterior, lateral, or posterior), supported by non-significant *p*-values and effect sizes under 0.1. At 1-year follow-up, 554 (91.6%) patients survived and 51 (8.4%) died. Their demographic, risk factors, and catheterization data are described in Table 2.

### 3.2. Denoised AP Signal

After implementing the proposed denoising method, we successfully extracted the clean segments of the AP signal. Figure 8 illustrates that the segments highlighted in red are the clearer, less noisy parts of the AP signal. These segments were then used to extract key features.

### 3.3. Extracted Features

A major challenge with the 18 features extracted initially (Table 1) was their high correlation. To address this, we calculated the correlation coefficient for each feature pair. We eliminated those with a correlation of more than 0.7, ensuring a more independent and effective set of features for our analysis. Without this correlation step, our feature selection methods might pick similar features and miss out on important different ones. The final set of features is nine in total, as shown in Table 3.

### 3.4. Feature Selection Results

#### 3.4.1. LBFS

After running our first feature selection method, which uses LDA 1000 times, based on how often each feature was chosen as the most sensitive one, we made a bar graph (Figure 9) to show how important each feature is. As illustrated in Figure 9, features like diastolic blood pressure (DBP), area under the curve (AOC), HB, skewness, and systolic blood pressure (SBP) consistently emerged as the top five selections in this method.

In a two-class problem with a single feature, LDA simplifies to a process of threshold determination. It starts by calculating the mean of each class’s single feature and then computes between-class and within-class variances. The main objective is to find an optimal threshold that maximizes the between-class variance relative to the within-class variance, effectively creating a decision boundary. This threshold helps classify new instances.

#### 3.4.2. PBFS

Using PCA as the foundation for our feature selection method, we generated a bar graph that highlights the most significant features identified through this approach. Figure 10 is the result of repeating the selection process 1000 times, showing which features are most important. The top five selected features were HB, SBP, FD, ascending time (AT), and skewness.

### 3.5. Results on ML Models (Hemodynamic Trace Features)

Table 4 shows mortality prediction using the five selected features from the AP signal. The performance of the predictions differs based on the classifier and the method used for selecting features. The accuracy of these predictions falls between 69% and 72%, while the AUC values range from 0.73 to 0.77.

The SVM classifier, employing PBFS, achieved the same levels of accuracy and specificity as the KNN classifier but showed a higher sensitivity. In comparison, the LDA classifier, utilizing PBFS features, achieved the best overall prediction, slightly outperforming both the KNN and SVM classifiers in AUC. When we used LBFS, the KNN classifier’s accuracy was lower, while LDA and SVM classifiers showed improved accuracy. The sensitivity and specificity across these classifiers also varied slightly. The most effective model among those tested was the LDA classifier.

Our method’s approach to handling data imbalance, which maintained the dataset’s distribution in our test set, resulted in an imbalanced test set. This led to a higher incidence of false positives compared to the use of a balanced test set, consequently yielding a lower precision value. Additionally, it is noteworthy that three out of the five features selected by each feature selection method remained the same, demonstrating consistency across different techniques.

### 3.6. Results from ML Models (Adding Demographics, Risk Factors, and Catheterization Data)

In addition to classification using features extracted only from the AP signal, we investigated whether adding extra information such as demographics, risk factors, and catheterization information to the top five features from the AP signal would improve the prediction accuracy. We chose these additional variables based on two criteria: their *p*-values and the sizes of their effects (details shown in Table 2). We selected variables with a *p*-value < 0.01 and an effect size > 0.2. The variables selected were age, renal dysfunction (RD), dialysis, and EST, with effect sizes of 0.57, 0.26, 0.24, and 0.46, respectively; all had a *p*-value < 0.01.

Table 5 shows the results of mortality prediction accuracy with the new variables added to the AP-driven features. By including these new features, we were able to keep the same level of sensitivity but we improved the accuracy, specificity, and AUC, particularly for the LDA classifier. Figure 11 displays the ROC curves for every trained model outlined in Table 5. The ROC curve and AUC values represent the average calculated across all folds.

## 4. Discussion

Using three supervised machine learning approaches (KNN, LDA, and SVM) for the first time, this study demonstrates that the AP-derived features can be effective in developing a risk-predictive model for 1-year outcome in patients with STEMI. When age, RD, dialysis, and EST information were also included, the accuracy and sensitivity of our best model (LDA with PBFS) improved by 7% and 9%, respectively, while maintaining almost the same specificity. For the same model, the AUC also improved from 0.77 to 0.82.

For denoising of the AP signal, we extracted characteristics primarily based on mean, SD, and FD. The mean and SD are utilized to establish a threshold for identifying and removing data segments with abnormally high or low AP waveform values. To the best of our knowledge, we are the first group to investigate the application of FD in the denoising of the AP signal. FD is a nonlinear metric that measures complexity and irregularity in time-domain signals. We used this feature to discard abnormally complex AP waveforms. In an existing study [46], researchers examined the influences of various signal properties such as amplitude, frequency, harmonics, noise power, and bandwidth on FD. They concluded that FD is effective for identifying structural changes in signals, providing a rapid and efficient way to assess variations in signal complexity.

We chose to use PBFS and LBFS methods to further reduce the number of features, allowing us to interpret the model better and enhance its predictive accuracy. Each method selected the top five features, providing more precise insights into the AP waveform characteristics associated with mortality. Figure 9 and Figure 10 display the ranked importance of each AP feature as determined by the LBFS and PBFS methods, respectively. We categorize important AP features into two groups. The first group of features are the overlapping features identified by both methods: skewness, HB, and SBP. The second group is the remaining two unique features selected by each method: DBP and AOC, chosen by LBFS, and AT and FD, selected by PBFS.

The features selected by our feature selection methods are supported by other studies which show their importance. In [47], entropy, skewness, and kurtosis values derived from ECG signals were fed into a least squares SVM classifier for MI detection. This research marks the first instance of calculating kurtosis and skewness from ECG signals specifically for mortality prediction, demonstrating their pivotal role in such predictive analysis. Similarly, previous research highlighted a positive correlation between elevated HB and mortality [48,49]. Lower SBP and DBP were also shown to be predictors for in-hospital, 30-day, and 1-year mortality in patients with STEMI [29]. Moreover, FD has been previously employed in the analysis of AP signals for various purposes, and it has an intricate association with arterial stiffness [50]; arterial stiffness is correlated with the risk of cardiovascular mortality and morbidity [51]. Abrupt myocardial damage due to STEMI compromises the myocardial ability to maintain adequate stroke volume or the amount of blood ejected per heartbeat. Many of these patients with low stroke volume are noted to have narrow pulse pressure (SBP–DBP, mmHg), slow uprise of the AP tracing (smaller AT), or smaller AOC [52,53].

In general, all classifiers, along with the selected features, yielded similar results. These results highlight the effectiveness of our method, utilizing only five features extracted from the AP signal. Notably, the LDA model with LBFS outperformed the others, achieving the highest accuracy, specificity, sensitivity, and AUC. Typically, machine learning models exhibit a slight decrease in performance on validation data (unseen data) compared to training data (seen data), as they are optimized based on the training dataset. In our study, this was true as well; the performance of the ML models on the validation dataset was slightly lower than on the training dataset, but it did not result in overfitting.

Incorporating demographics, risk factors, and catheterization information such as age, RD, dialysis, and EST into the selected features led to improvements across various metrics. The most effective model, which demonstrated a slightly better AUC, was the LDA trained with PBFS. Both selected risk factors are related to abnormal kidney function, emphasizing its crucial role. The relationship between renal dysfunction and cardiovascular outcomes in the general population [54], and STEMI patients [55] is well established. Age has been widely used as a predictor for mortality, especially in-hospital [29,56], 30-day [29], and 1-year mortality [29,57,58]. Similarly, EST is also independently linked to a higher risk of cardiovascular disease and related death [53].

Different studies have previously focused on mortality prediction in patients with STEMI using different methodologies. Oliveira et al. [27] conducted a study employing ML algorithms to predict in-hospital mortality in acute MI patients, including STEMI cases. Their research involved three distinct experiments, each utilizing varying feature sets. The first two experiments used 1179 discharge episodes, initially focusing on admission variables and adding laboratory data, comorbidities, and interventions. The third experiment, using 445 episodes, included more specific pathology-related variables than the previously added variables. The best performance was observed in the third experiment, without data balancing and with all 44 variables, where the KNN algorithm achieved 87% accuracy, 36% precision, 90% recall, and an AUC of 0.89. In another study [28] involving 3191 STEMI patients, five different machine learning models were trained and tested using 31 candidate features, with the Extra-tree classifier proving to be the most effective for predicting all-cause 30-day mortality following STEMI. This model achieved a sensitivity of 85%, specificity of 74%, accuracy of 79%, and an AUC of 79.7%. In another study [29] using the data from a registry of 27592 STEMI patients, researchers applied ML to predict and identify factors associated with short- and long-term mortality in Asian patients with STEMI. These models were developed for in-hospital (6299 patients), 30-day (3130 patients), and 1-year (2939 patients) mortality. The analysis considered 50 variables (9 continuous, 41 categorical) and three ML algorithms (RF, SVM, and Linear Regression). This study evaluated model performance using both a complete and reduced set of variables, achieving an AUC ranging from 0.73 to 0.90. SVM classifier (with feature selection) displayed the highest predictive performance for in-hospital, 30-day, and 1-year models, achieving AUCs of 0.88, 0.90, and 0.84, respectively. Notably, for 1-year mortality prediction the same model achieved the best results with an accuracy of 77%, specificity of 77%, and sensitivity of 75%.

Our study has several limitations. First, the patient data is collected from a single institution, which may not be universally representative and could introduce bias in outcome measurements. However, our hospital is the only cardiac center providing tertiary cardiac care in the province of Manitoba. Second, not all potential risk factors leading to STEMI were available or included in this study. Thirdly, the limited number of patients is a major limitation and expanding the dataset could lead to an increased number of patients in the non-survivor group, potentially enhancing prediction. Fourthly, we could not compare our results with conventional scoring methods due to a lack of access to Killip class data that is essential for conventional scoring methods [59]. Finally, another challenge was our dataset’s highly imbalanced nature. We developed a method to address this imbalance while retaining key features and preserving the dataset’s distribution. Although this strategy improved our model training, the imbalance inevitably may affect the observed predictions. One notable consequence is reduced precision, resulting from maintaining class distribution in our final test set. This may have led to a higher representation of patients in class 1 (survived) compared to class 2 (the non-survived group), resulting in an increased number of false positives and a consequent significant decrease in precision.

In our future research, we plan to extract additional features from the AP signal to develop more comprehensive models. The dicrotic notch is a key point in AP. It identifies a specific point on the AP curve, allowing us to divide the waveform into systolic and diastolic sections. This distinction is vital as it enables us to extract different features from these two phases of heart function, compare them, and potentially utilize them in training our advanced models. However, accurately identifying the dicrotic notch can be challenging, as it may not be present in all waveforms and may require varied strategies for different waveforms or patients. A more complex model we aim to develop involves tree-based methods, which have shown effectiveness in mortality prediction in other studies [32,60,61]. These methods offer promising avenues for enhancing our predictive capabilities. While our primary focus in this study was on predicting mortality at 1-year post-PPCI, future research could expand to include other complications, including but not limited to prolonged in-hospital stays, identifying new diagnoses of heart failure, and more. This broader scope could provide deeper insights into identifying high-risk patients, which is important as careful monitoring and timely intervention can plausibly improve outcomes and quality of life and reduce health-related expenditure.

## 5. Conclusions

Machine learning analyzing AP signal, incorporating other clinical parameters, can predict 1-year mortality in STEMI patients treated with PPCI. Our work showed that such a hemodynamic tracing has the potential to be a marker of clinical significance in identifying patients at risk for adverse outcomes. We identified skewness, HB, and SBP as the most significant AP features for our prediction. Our findings should be validated in a larger, prospective, multi-center study.

## Figures and Tables

**Figure 1 medicina-60-00558-f001:**
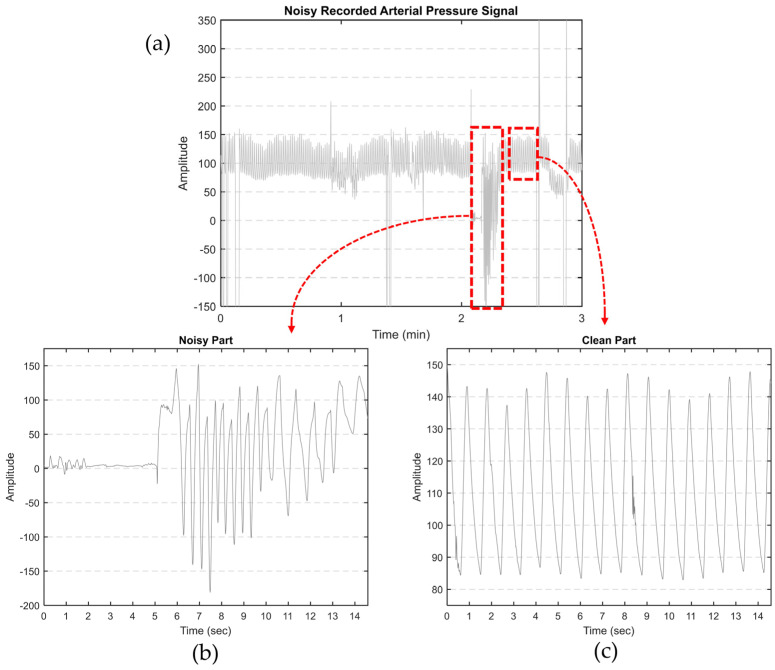
(**a**) Noisy AP signal recorded by MacLab. The artifacts were manually clipped to the maximum shown in the figure; (**b**) an example of noisy and (**c**) clean parts of a typical recorded AP signal.

**Figure 2 medicina-60-00558-f002:**
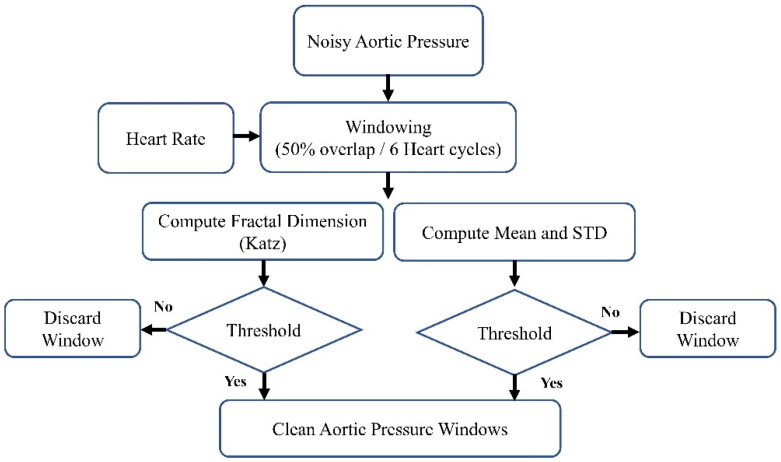
The block diagram of the denoising method.

**Figure 3 medicina-60-00558-f003:**
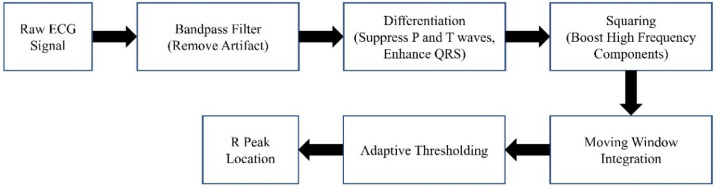
The Pan–Tompkins algorithm.

**Figure 4 medicina-60-00558-f004:**
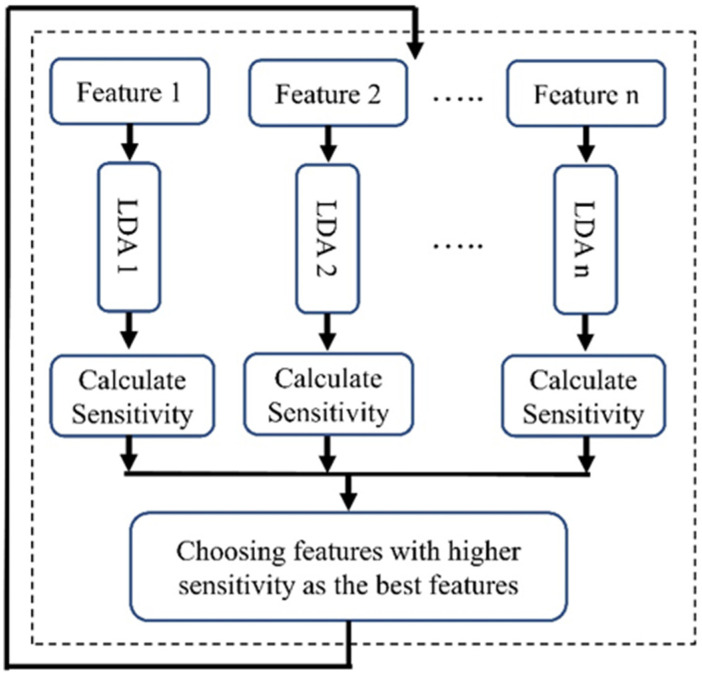
Block diagram of the LBFS method.

**Figure 5 medicina-60-00558-f005:**
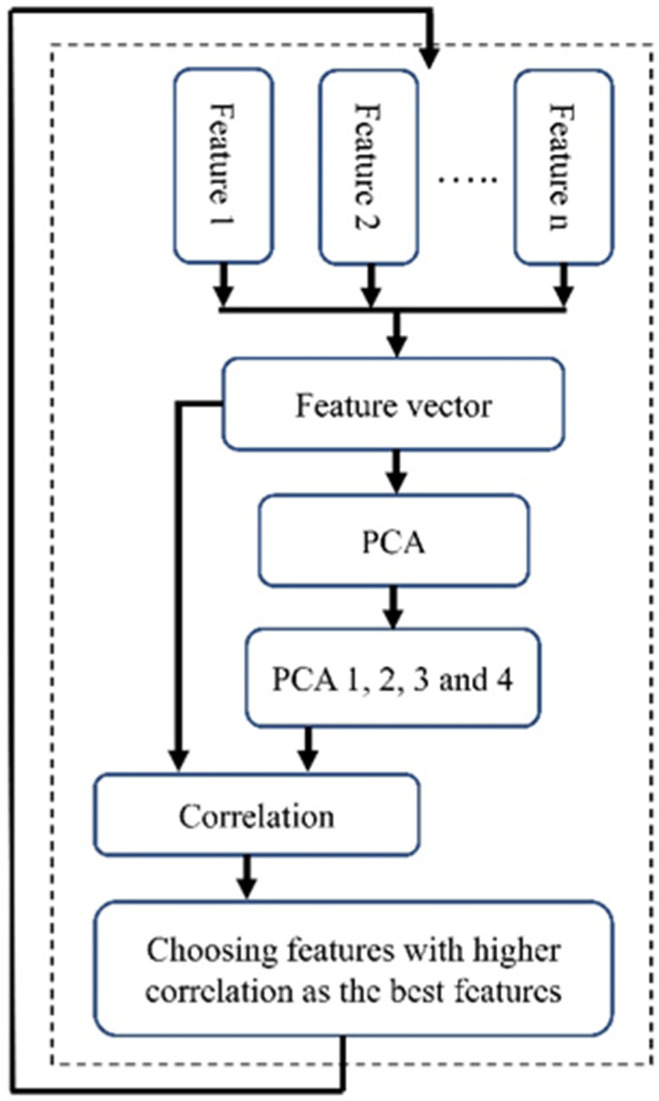
The block diagram of the PBFS method.

**Figure 6 medicina-60-00558-f006:**
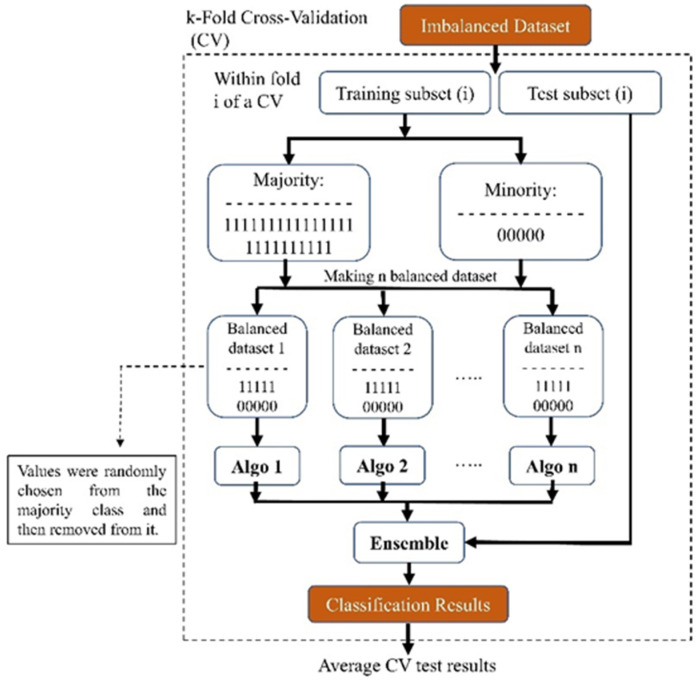
Block diagram of our proposed ensemble-based method for addressing imbalanced datasets.

**Figure 7 medicina-60-00558-f007:**
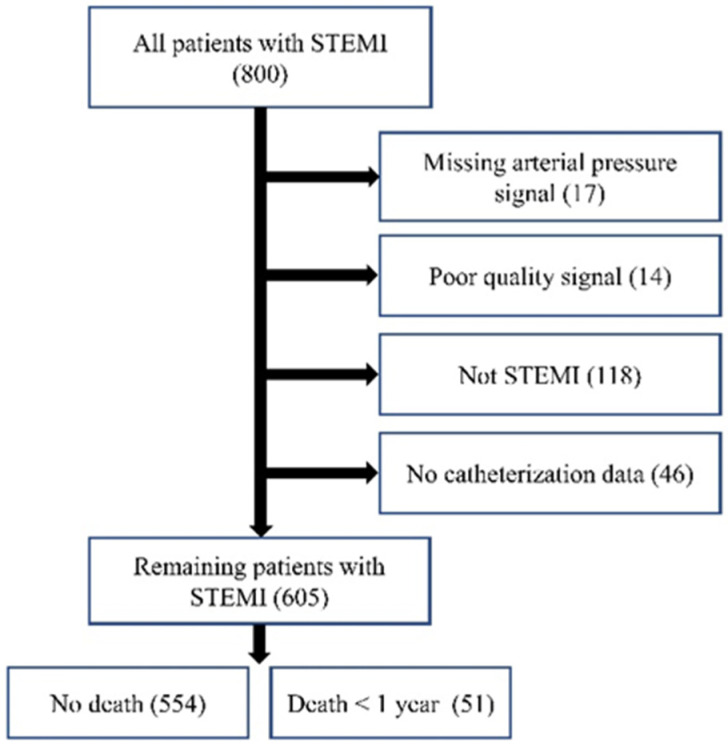
Data collection flowchart.

**Figure 8 medicina-60-00558-f008:**
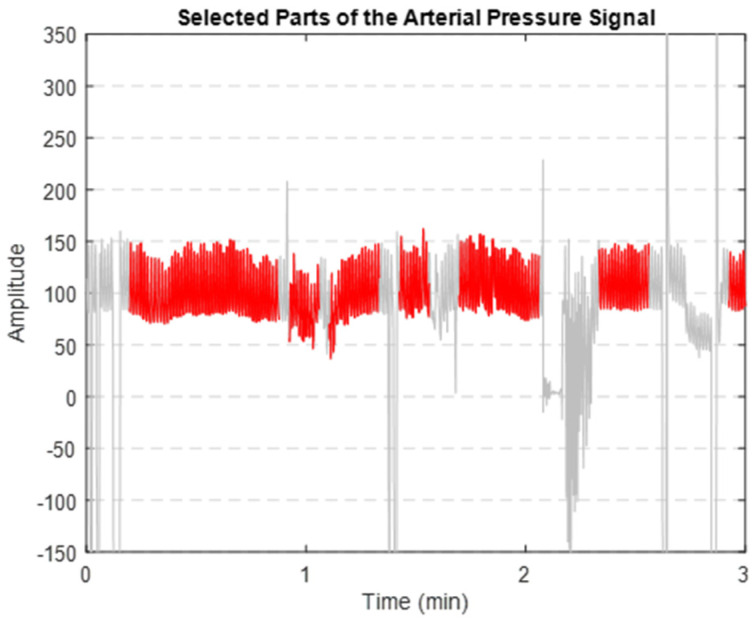
Extracted clean segments (shown in red) of the AP signal (depicted in grey).

**Figure 9 medicina-60-00558-f009:**
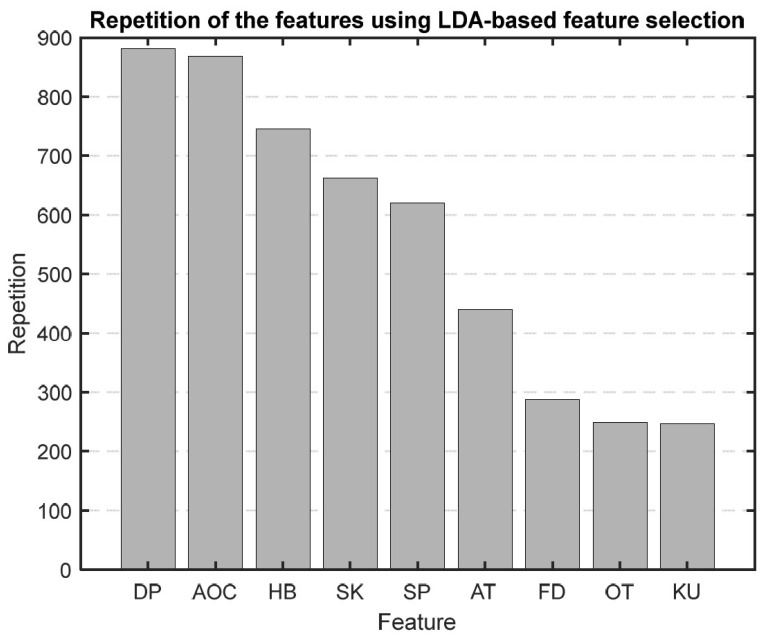
Bar graph showing the repetition of each feature in the LBFS method.

**Figure 10 medicina-60-00558-f010:**
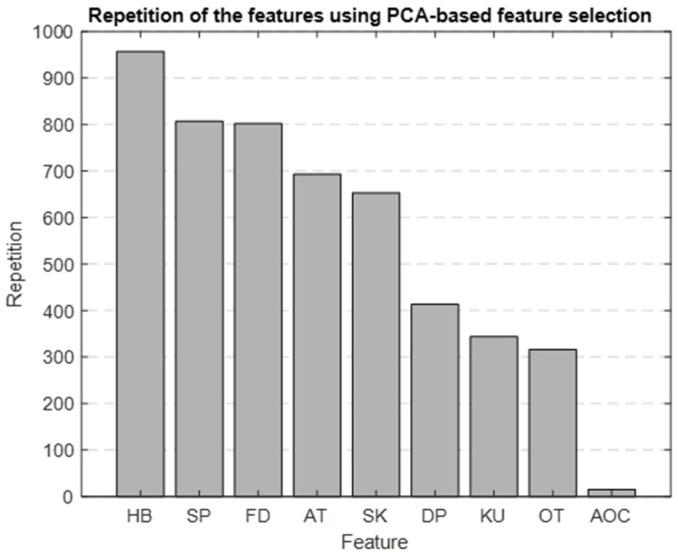
Bar graph showing the repetition of each feature in the PBFS method.

**Figure 11 medicina-60-00558-f011:**
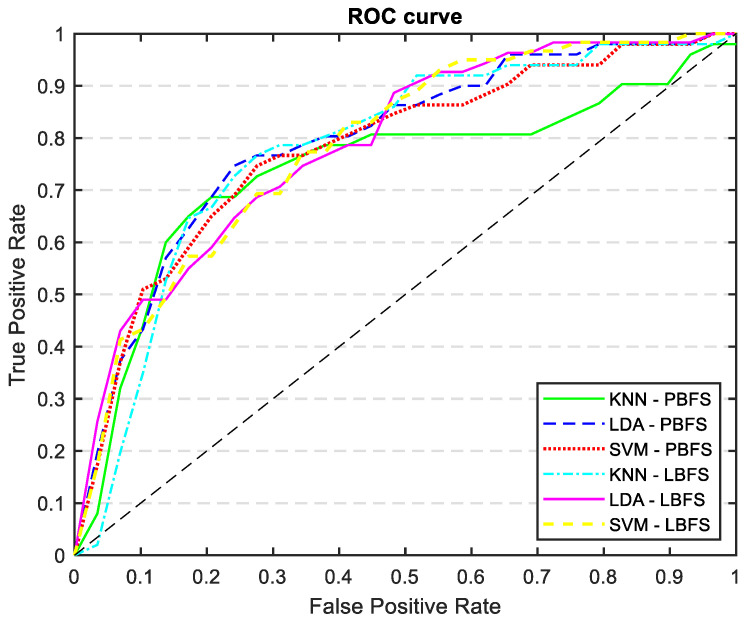
ROC curves of all the trained models: KNN, LDA, and SVM classifiers trained with PCA and LDA selected features, incorporating demographics, risk factors, and catheterization information.

**Table 1 medicina-60-00558-t001:** The extracted 18 features.

	Feature	Abbreviation	Definition	*p*-Value (Death < 1 Year vs. No Death)
1	Heartbeat	HB	60/(*t* @ end)	<0.01
2	Diastolic blood pressure	DBP	Min (p)	<0.01
3	Systolic blood pressure	SBP	Max (p)	<0.01
4	Pulse pressure	PP	SBP—DBP	<0.01
5	Mean arterial pressure	MAP	(2 × DBP + SBP/3	<0.01
6	Overall time	OT	Whole time of the surgery	<0.01
7	Ascending time	AT	*t* @ systolic peak	<0.01
8	Descending time	DT	(*t* @ end)—AT	<0.01
9	Total area under the curve	AOC	∫p	<0.01
10	Ascending area	UA	∫p (Ascending portion)	<0.01
11	Descending area	DA	AOC—UA	<0.01
12	Area Ratio	AR	DA/UA	<0.01
13	Maximum slope	MS	$\mathrm{Max}\;(p^{\prime })$	<0.01
14	Fractal dimension	FD	Kats FD method	<0.01
15	Skewness	SK	$E{\left(p-\mu \right)}^{3}/\sigma^{3}$	<0.01
16	Kurtosis	KU	$E{\left(p-\mu \right)}^{4}/\sigma^{4}$	<0.01
17	Spectral Entropy	SE	−Sum (PSD × log2(PSD))	<0.01
18	Average power	Pave	Sum (PSD)/*n*	<0.01

*p*-value derived from Mann–Whitney U test. p is a single AP waveform, $p^{\prime }$ is the first derivation of the waveform, *t* is time, *n* is the number of samples in the waveform, *μ* is the mean of samples of the waveform, *σ* is the standard deviation, and PSD is power spectral density.

**Table 2 medicina-60-00558-t002:** Demographic, risk factors, and catheterization data for STEMI patients of this study.

Characteristics	Total	No. Death	Death < 1 Year	*p*-Value (Death < 1 Year vs. No. Death)	Effect Size
No. of Patients	605	554 (91.57%)	51 (8.43%)		
Demographics					
Age, years	64.19 ± 13.18	63.56 ± 12.10	70.96 ± 13.38	<0.001	0.57
Male, no. (%)	432 (71.41%)	405 (73.10%)	27 (52.94%)	0.007	0.13
Weight, kg	171.65 ± 10.76	171.93 ± 10.68	168.73 ± 11.27	0.016	0.30
Height, cm	85.64 ± 20.13	86.24 ± 19.99	79.15 ± 20.69	0.043	0.35
BMI, kg/m^2^	29.14 ± 8.58	29.27 ± 8.76	27.7 ± 6.29	0.12	0.18
Risk Factors, no. (%)					
Hypertension	349 (57.67%)	314 (56.68%)	35 (68.63%)	0.098	0.07
DM	156 (25.79%)	139 (25.09%)	17 (33.33%)	0.19	0.05
Dyslipidemia	246 (40.66%)	230 (41.52%)	16 (31.37%)	0.16	0.06
Stroke or TIA	28 (4.63%)	21 (3.79%)	7 (13.72%)	0.001	0.13
PVD	19 (3.14%)	15 (2.71%)	4 (7.84%)	0.044	0.08
RD	38 (6.28%)	24 (4.33%)	14 (27.45%)	<0.001	0.26
Dialysis	5 (0.83%)	1 (0.18%)	4 (7.84%)	<0.001	0.24
History of IHD	116 (19.17%)	108 (19.49%)	8 (15.68%)	0.51	0.03
PCI or CABG	90 (14.88%)	85 (15.34%)	5 (9.8%)	0.29	0.04
Catheterization Data					
ESP, s/min	19.11 ± 3.29	19.19 ± 3.23	18.31 ± 3.82	0.042	0.27
EST, s/beat	0.24 ± 0.04	0.24 ± 0.04	0.22 ± 0.05	0.002	0.46

Continuous variables are shown as mean ± SD. Categorical variables are shown as portions of the group. *p*-values were derived from Mann–Whitney U and χ2 (chi-square) tests for continuous and categorical variables, respectively. BMI—body mass index, DM—diabetes mellitus, TIA—transient ischemic attack, PVD—peripheral vascular disease, RD—renal dysfunction, IHD—ischemic heart disease, PCI—percutaneous coronary intervention, CABG—coronary artery bypass graft, ESP—ejection systolic period, and EST—ejection systolic time.

**Table 3 medicina-60-00558-t003:** The extracted features after removal of highly correlated features.

	Feature	Abbreviation	Definition	*p*-Value (Death < 1 Year vs. No. Death)
1	Heartbeat	HB	60/(*t* @ end)	<0.001
2	Diastolic blood pressure	DBP	min (p)	<0.001
3	Systolic blood pressure	SBP	max (p)	<0.001
4	Overall time	OT	Whole time of the surgery	<0.001
5	Ascending time	AT	*t* @ systolic peak	<0.001
6	Total area under the curve	AOC	∫p	<0.001
7	Fractal dimension	FD	Kats FD method	<0.001
8	Skewness	SK	$E{\left(p-\mu \right)}^{3}/\sigma^{3}$	<0.001
9	Kurtosis	KU	$E{\left(p-\mu \right)}^{4}/\sigma^{4}$	<0.001

*p*-value derived from Mann–Whitney U test. p is a single AP waveform, *t* is time, *μ* is the mean of samples of the waveform, *σ* is the standard deviation, and PSD is power spectral density.

**Table 4 medicina-60-00558-t004:** Prediction results using three different classifiers with features selected by PBFS and LBFS methods.

	PBFS			LBFS		
Classifier	KNN	LDA	SVM	KNN	LDA	SVM
Accuracy (%)	70	71	70	69	72	72
Specificity (%)	70	71	70	69	72	73
Sensitivity (%)	69	70	72	73	73	68
Precision (%)	18	18	18	18	19	19
AUC	0.73	0.77	0.76	0.73	0.77	0.74

**Table 5 medicina-60-00558-t005:** Prediction results after adding demographic, risk factors, and catheterization information.

	PBFS			LBFS		
Classifier	KNN	LDA	SVM	KNN	LDA	SVM
Accuracy (%)	78	78	77	79	76	77
Specificity (%)	79	80	77	80	76	78
Sensitivity (%)	73	71	73	71	67	68
Precision (%)	23	24	23	25	21	22
AUC	0.76	0.82	0.81	0.81	0.81	0.81

## Data Availability

The datasets presented in this article are not readily available as the data are part of an ongoing study. Requests for accessing the dataset should be directed to Dr. Ashish H. Shah.

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
