# Peer review of "1-Year Mortality Prediction through Artificial Intelligence Using Hemodynamic Trace Analysis among Patients with ST Elevation Myocardial Infarction"

_medicina, 2024, doi:10.3390/medicina60040558_

Round 1

Reviewer 1 Report

Comments and Suggestions for Authors

Overall, the manuscript presents a promising study aimed at improving risk prediction for patients undergoing PPCI in STEMI.  With some minor revisions to improve clarity and provide additional context, it has the potential to make a substantial contribution to the literature on risk prediction in STEMI patients undergoing PPCI.

1. In table 1, use p<0.001 rather than ****=p<0.0001. Also, table 2. What is the benefit of classifying the three (uncommon) level of significant p values?

2. Despite providing detailed appropriate methodology, ot lacks citations regarding ML algorithms etc.

Author Response

Comments to Author:

In table 1, use p<0.001 rather than ****=p<0.0001. Also, table 2. What is the benefit of classifying the three (uncommon) level of significant p values?

Reply: We thank the reviewer for the comment. We completely agree with your observations and have accordingly revised the p-values in Tables 1 and 3. Additionally, we have omitted the notation "****=p<0.0001" from the footnotes of these tables.

The reason for having four different levels for p-values (*, **, ***, and ****) was to facilitate a quick and clear understanding of the impact of each risk factor by quick look at the table. However, recognizing your point that this practice is uncommon, we have updated Table 2 accordingly. Now, for p-values greater than 0.001, we explicitly display the values. For p-values less than 0.001, we indicate them as "p<0.001".

We have also revised the notation for p-value levels in Table 2 and removed the detailed legend denoting "NS=not significant, *=p<0.05, **=p<0.01, ***=p<0.001, ****=p<0.0001" from the table's footnote.

Comments to Author:

Despite providing detailed appropriate methodology, it lacks citations regarding ML algorithms etc.

Reply: We thank the reviewer for the comment. In methodology, we referred to ML algorithms in two sections:

In section 2.5, when we first discussed the PCA and LDA methods, it was noted that citations for these two algorithms were not included. The revised paragraph, which is also highlighted in the revised paper, is as follows:

Line  202 – 205:

We also implemented feature selection methods to reduce the number of variables further, thereby shortening training time and enhancing model performance. We employed two feature reduction methods: principal component analysis (PCA) [40] and linear discriminant analysis (LDA) [41]. In both approaches, we selected the top five features.

In section 2.7, where we discussed the three ML algorithms used for binary classification, the proper citations for the three utilized ML models are provided. They are as follows:

Line  279 – 281:

The prediction models for 1-year mortality were developed using three different machine learning techniques: K-nearest neighbor (KNN) [43], LDA [41], and support vector machine (SVM) [44].

Reviewer 2 Report

Comments and Suggestions for Authors

The article entititled "One-year Mortality Prediction through Artificial Intelligence using Hemodynamic Trace Analysis among Patients with ST Elevation Myocardial Infarction" is very interesting and well described. However, minor revisions could further be neccesary. 

1. The authors should discuss mor about risk assessments by incorporating additional scoring systems beyond GRACE and TIMI. Moreover, a comparison between the conventional scoring methods and new technologies, such as machine learning would be necessary. 

2. The distribution of STEMI myocardial infarction on different regions, such as inferior, anterior, lateral, and posterior, may have a prognostic outcomes. More than this, the patients with multiple coronary localizations could influence the final score and are needed to be discussed.

3. The inclusion of future research in the Conclusion section is not properly discussed. Those information would be more suitably integrated in the discussion section. It would be better if the conclusion are extended with the personal study’s findings, not just the generalized statements.

Comments on the Quality of English Language

English language is fine, just minor revision.

Author Response

We are incredibly grateful to the Reviewers for their thorough reviews. The reviews contain valuable comments, which have helped us clarify our paper. The manuscript has been revised according to the reviewers’ comments.  A point-by-point reply is given below.  The line, figure and table numbers noted in the replies refer to the revised manuscript.

In the following response, the reviewers’ comments are shown in black text, and our responses appear immediately below in blue text.

Response to Reviewer #2:

Comments to Author:

The authors should discuss more about risk assessments by incorporating additional scoring systems beyond GRACE and TIMI. Moreover, a comparison between the conventional scoring methods and new technologies, such as machine learning, would be necessary.

Reply: We thank the reviewer for the comment. We have updated the introduction. We discussed more about risk assessments and added additional systems.

Line  66 – 77:

Various risk assessments in the context of myocardial infarction (MI) have been developed. The Global Registry of Acute Coronary Events (GRACE) [15], the most widely used and recommended by the European Society of Cardiology STEMI guidelines [16], estimates the mortality risk in hospital, at 6 months, 1 year, and 3 years. The thrombolysis in myocardial infarction (TIMI) [17] risk score was initially developed for 30-day mortality in patients after thrombolysis and then validated for patients with STEMI [18]. Based on clinical and electrocardiographic characteristics, the primary angioplasty in myocardial infarction (PAMI) score is used to predict late mortality in patients with STEMI treated by PPCI [19]. The controlled abciximab and device investigation to lower late angioplasty complications (CADILLAC) considers the initial measurement of left ventricular function and predicts 1-year mortality [20]. Finally, Zwolle [21] score was developed for 30-day mortality prediction.

We discussed about the differences between conventional scoring method and ML:

Line  78 – 93:

These traditional methods for determining cardiovascular disease (CVD) risk typically presuppose a linear relationship between risk factors and clinical outcomes. However, such a linear approach might be oversimplifying their relationship. Cardio-vascular diseases are inherently complex and diverse, influenced by genetic predispositions, environmental conditions, and lifestyle choices [16,22]. These approaches primarily focus on conventional prognostic factors [23], limiting their effectiveness due to the emerging need to incorporate and examine various information sources, including those describing MI-related pathophysiology. Moreover, these scoring systems are routinely not utilized in the current era of prompt coronary revascularization. Aortic pulse wave is a physiological marker describing cardiovascular health [24,25] and may provide valuable information about changing physiological status among patients undergoing PPCI.

Machine learning (ML) has the potential to bypass the restrictions of the approaches mentioned above [26]. Static assumptions about data behaviour do not con-strain ML data analysis and can train models to uncover patterns within the data. The application of ML, especially in predicting in-hospital mortality [27], 30-day mortality [28], short- and long-term mortality [29], arrhythmia [30] and readmission [31], has seen rapid growth.

We also revised the following sentences for clarifying and better comparison:

Line  93 – 99:

ML has been widely compared with traditional methods such as TIMI and GRACE. ML has demonstrated superior performance to traditional risk-scoring methods in mortality prediction [32–36]. ML outperformed in predicting both short- and long-term mortality compared to the TIMI score [33]. Additionally, it demonstrated better outcomes for 30-day [35] and 1-year [32,34,36] mortality predictions compared to the GRACE score for patients with STEMI.

All the above-mentioned scoring systems need Killip class for calculation, and we did not have access to that. So, we included this sentence in the limitation section:

Line  493 – 495:

Fourthly, we could not compare our results with conventional scoring methods due to lack of access to Killip class data that is essential for conventional scoring methods [59].

Comments to Author:

The distribution of STEMI myocardial infarction on different regions, such as inferior, anterior, lateral, and posterior, may have a prognostic outcome. More than this, the patients with multiple coronary localizations could influence the final score and are needed to be discussed.

Reply: We thank the reviewer for the comment. We analysed the relation between each STEMI type and the outcome (1-year mortality). As we can see in the following table, the p-values were not significant, and the effect sizes are small. So, we excluded STEMI types from further analysis.

Characteristics

Total

No death

Death < 1 year

P-value

(Death <1 year vs. No death)

Effect size

No. of patients

605

554 (91.57%)

51 (8.43%)

STEMI type

Inferior, n(%)

313 (51.74%)

289 (52.17%)

24 (47.06%)

0.48

0.028

Anterior, n (%)

230 (38.02%)

210 (37.91%)

20 (39.22%)

0.85

0.008

Lateral, n(%)

43 (7.14%)

38 (6.86%)

5 (9.80%)

0.43

0.032

Posterior, n(%)

19 (3.14%)

17 (3.07%)

2 (3.92%)

0.74

0.014

We added the following sentence in line 315 - 319:

We investigated how the site of STEMI impacts patient survival, differentiating be-tween survivors and non-survivors. Our research found no significant connection be-tween survival rates and the STEMI's location (inferior, anterior, lateral, or posterior), supported by non-significant p-values and effect sizes under 0.1.

We collected the STEMI type from MacLab, which is documented by the doctor during the revascularization. Each patient has one type in MacLab, and it can be inferior, anterior, lateral, or posterior. Having multiple coronary localizations has not been directly included in our analysis. However, we included the time of the surgery as a feature in our analysis. A long procedural time indicates a more complicated procedure. Patients with such challenging and complex procedures may experience adverse outcomes. 

As we can see in Figures 9 and 10, overall time (OT) was not selected by our feature selection methods as a key feature for mortality prediction. For this reason, we will not further discuss this feature in the discussion section.

Although, as we mentioned, we did not directly include having multiple coronary localizations as the key predictor in our analysis.

Comments to Author:

The inclusion of future research in the Conclusion section is not properly discussed. Those information would be more suitably integrated in the discussion section. It would be better if the conclusion are extended with the personal study’s findings, not just the generalized statements.

Reply: We thank the reviewer for the comment. We moved the future research to the end of the discussion.

We also changed the conclusion section slightly:

Machine learning analysing AP signal, incorporating other clinical parameters can predict 1-year mortality in STEMI patients treated with PPCI. Our work showed that such a hemodynamic tracing has potential to be a marker of clinical significance in identifying patients at risk for adverse outcomes. We identified skewness, HB, and SBP as the most significant AP features for our prediction. Our findings should be validated in a larger, prospective, multi-centre study.

Comments to Author:

English language is fine, just minor revision

Reply: We thank the reviewer for the comment. We have checked the manuscript and revised it entirely.
